# Occurrence of long-term depression in the cerebellar flocculus during adaptation of optokinetic response

**Takuma Inoshita, Tomoo Hirano\***

Department of Biophysics, Graduate School of Science, Kyoto University, Sakyo-ku, Japan

**Abstract** Long-term depression (LTD) at parallel fiber (PF) to Purkinje cell (PC) synapses has been considered as a main cellular mechanism for motor learning. However, the necessity of LTD for motor learning was challenged by demonstration of normal motor learning in the LTD-defective animals. Here, we addressed possible involvement of LTD in motor learning by examining whether LTD occurs during motor learning in the wild-type mice. As a model of motor learning, adaptation of optokinetic response (OKR) was used. OKR is a type of reflex eye movement to suppress blur of visual image during animal motion. OKR shows adaptive change during continuous optokinetic stimulation, which is regulated by the cerebellar flocculus. After OKR adaptation, amplitudes of quantal excitatory postsynaptic currents at PF-PC synapses were decreased, and induction of LTD was suppressed in the flocculus. These results suggest that LTD occurs at PF-PC synapses during OKR adaptation.
DOI: https://doi.org/10.7554/eLife.36209.001

## Introduction

The cerebellum plays a critical role in motor learning, and a type of synaptic plasticity long-term depression (LTD) at parallel fiber (PF) to Purkinje cell (PC) synapses has been considered as a primary cellular mechanism for motor learning (*Ito, 1989*; *Hirano, 2013*). However, the hypothesis that LTD is indispensable for motor learning was challenged by demonstration of normal motor learning in rats in which LTD was suppressed pharmacologically or in the LTD-deficient transgenic mice (*Welsh et al., 2005*; *Schonewille et al., 2011*). More recently, LTD induction by different conditioning stimulation in the above transgenic mice was reported (*Yamaguchi et al., 2016*), and the ideas that multiple types of synaptic plasticity in the cerebellum contribute to motor learning, and that some types of plasticity might compensate defects of the other have been proposed (*Boyden et al., 2004*; *Dean et al., 2010*; *Gao et al., 2012*; *Hirano, 2013*).

Most of previous studies have addressed roles of LTD in motor learning by examining effects of LTD-suppression or LTD-facilitation on the motor learning ability (*Aiba et al., 1994*; *De Zeeuw et al., 1998*; *Hirai et al., 2005*; *Hansel et al., 2006*; *Takeuchi et al., 2008*). Here, we took a different approach, and asked whether LTD occurs during a motor learning process or not in wild-type animals, which has been rarely examined. An exception was the study by *Schreurs et al. (1997)* which reported suppression of LTD at PF-PC synapses after classical conditioning in rabbits, although whether amplitudes of synaptic responses were depressed was not examined. We also note that recent study by *Nguyen-Vu et al. (2017)* suggested occurrence of LTD saturation in double knockout mice of MHC1 H2-K$^b$/H2-D$^b$ with enhanced LTD.

Here, we used adaptation of optokinetic response (OKR) as a model of cerebellum-dependent motor learning and examined how it influenced LTD. OKR is a reflexive eye movement following motion of the large visual field to suppress blur of visual image during animal movement. The

**\*For correspondence:**
thirano@neurosci.biophys.kyoto-u.ac.jp

**Competing interests:** The authors declare that no competing interests exist.

efficacy of OKR can be increased by applying continuous movement of the large visual field, which is called OKR adaptation. Previous studies reported that the horizontal (H) zone of flocculus, a small region of cerebellum, is necessary for adaptation of horizontal OKR (*Robinson, 1981*; *Nagao, 1988*; *du Lac et al., 1995*; *Boyden et al., 2004*). Thus, we have examined changes of PF-PC synaptic transmission in the cerebellar flocculus, which was induced during OKR adaptation.

## Results

### Adaptation of optokinetic response

We recorded OKR in a mouse by applying optokinetic stimulation, which was horizontal sinusoidal rotation of a screen with vertical black and white stripes (14°) at 1 Hz, 10°/sec peak velocity (*Wakita et al., 2017*). Continuous optokinetic stimulation (OKR training, 50 s stimulations with 10 s intervals, repeated for 60 min) enhanced the eyeball movement. The gain of OKR which was defined as the amplitude of eye movement divided by the screen movement, was significantly increased by 1 hr OKR training (before, 0.32 ± 0.01; after, 0.62 ± 0.02, n = 31, p<0.001, paired t-test) (*Figure 1*).

### Smaller unitary excitatory postsynaptic response in the H zone of OKR trained mice

If LTD occurs during OKR training, the synaptic transmission at PF-PC synapses should be depressed. LTD is expressed at the postsynaptic side of PF-PC synapses (*Hirano, 1991*; *Linden, 2001*). Thus, the amplitude of unitary postsynaptic response in a PC might be decreased by the OKR training. Previous studies showed that unitary synaptic response caused by a single synaptic vesicle can be recorded as an asynchronous quantal excitatory postsynaptic current (qEPSC) in the $Sr^{2+}$ containing extracellular solution (*Li et al., 1995*; *Abdul-Ghani et al., 1996*). We recorded PF-specific qEPSCs induced by stimulation of PFs in the molecular layer of cerebellar slices prepared from trained or untrained mice (*Figure 2*). The amplitude of qEPSCs recorded in the lateral half of H zone in the flocculus was significantly smaller in the trained mice (9.73 ± 0.12 pA, n = 16) than in the untrained mice (10.58 ± 0.24 pA, n = 14, p=0.003, Student's t-test). These results include the data obtained in blind experiments (trained, 9.81 ± 0.15 pA, n = 5; untrained 10.50 ± 0.23 pA, n = 5, p=0.04, Student's t-test). We note that only qEPSCs > 6 pA were analyzed (*Figure 2C,D*). Thus, the mean amplitudes should have been overestimated, and the overestimation should have been larger when there were more small events. Therefore, the real difference of qEPSC amplitudes between

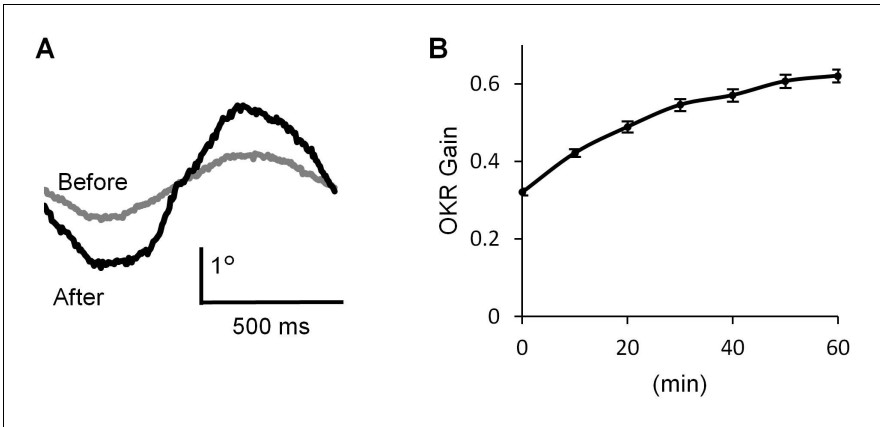

**Figure 1.** OKR adaptation. (**A**) Representative eye position traces before (grey) and after (black) 60 min continuous optokinetic stimulation (OKR training). Ten traces were averaged for each. (**B**) OKR gain changes during the OKR training (n = 31). Mean ± SEM.
DOI: https://doi.org/10.7554/eLife.36209.002
The following source data is available for figure 1:

**Source data 1.** Time course of OKR gain change during OKR training.
DOI: https://doi.org/10.7554/eLife.36209.003

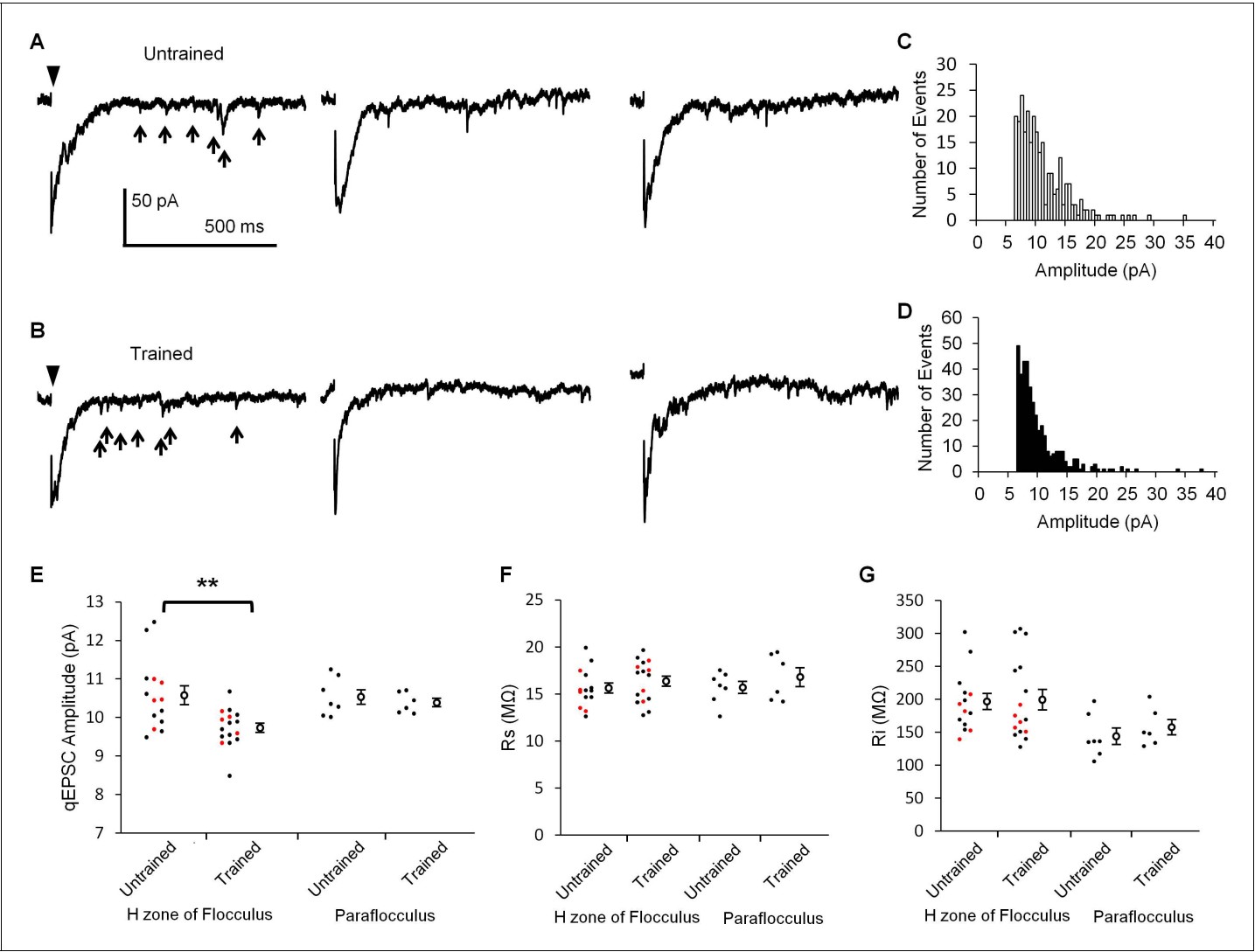

**Figure 2.** OKR training decreased qEPSC amplitudes. (**A, B**) Representative current traces showing qEPSCs (arrows) induced by parallel fiber stimulation (arrowheads) in the H zone of untrained (**A**) or trained mice (**B**). (**C, D**) Histograms of qEPSC amplitudes, which were obtained from the same PC as shown in (**A**) or (**B**). (**E**) Distribution of mean qEPSC amplitudes. Each small circle represents the mean amplitude in a PC (red, blind experiment; black, non-blind experiment), and large open circles and bars represent Mean ± SEM of all data. (**F, G**) Series (**F**) and input (**G**) resistances of PCs. Data shown in (**E, F, G**) were recorded with or without OKR training in both the flocculus and paraflocculus. \*\*p<0.01, Student's t-test.
DOI: https://doi.org/10.7554/eLife.36209.004

The following source data is available for figure 2:

**Source data 1.** qEPSC amplitudes with or without OKR training in the H-zone of flocculus or in the paraflocculus.
DOI: https://doi.org/10.7554/eLife.36209.005

with and without OKR training might have been larger. We also recorded qEPSCs in the dorsal part of paraflocculus, which was a cerebellar region located next to the flocculus and had little involvement in OKR adaptation (*Ito et al., 1982*; *Nagao, 1989*; *Wang et al., 2014*). The qEPSC amplitudes were not significantly different between the trained (10.39 ± 0.11 pA, n = 6) and the untrained mice (10.53 ± 0.19 pA, n = 7, p=0.53, Student's t-test). The series and input resistances were not different between with and without OKR training in both the flocculus and paraflocculus (*Figure 2F,G*). Thus, the continuous optokinetic stimulation inducing OKR adaptation specifically depressed the postsynaptic responsiveness of PF-PC synapses in the H zone of flocculus, supporting the idea that LTD takes place during the OKR training.

## Suppression of LTD induction in the H zone of trained mice

If depression of PF-PC synaptic transmission reflects the LTD expression during the OKR training, further induction of LTD might be affected. To test this possibility, we next examined LTD induction after the OKR training. Application of depolarization of a PC (0 mV, 50 ms, 1 Hz, 10 times) coupled with the presynaptic PF stimulation at 15 ms after the onset of PC depolarization (*Tanaka et al., 2013*), induced LTD at PF-PC synapses in the H zone of untrained mice (30 min, 69.5 ± 3.4%, n = 14, p<0.001, paired t-test) (*Figure 3*), whereas the same conditioning stimulation induced weak LTD in the trained mice (97.0 ± 2.6%, n = 12, p=0.03, paired t-test). The decrease in EPSC amplitude at 30 min after the conditioning stimulation was significantly smaller in the trained mice than in the untrained mice (p<0.001, Student's t-test). These results include the data obtained in blind experiments (trained, 97.4 ± 4.0%, n = 5; untrained, 72.6 ± 5.5%, n = 6, p=0.007, Student's t-test). On the other hand, in the paraflocculus OKR training did not affect the LTD induction (untrained, 71.6 ± 5.0%, n = 6, p=0.004; trained, 75.6 ± 4.6%, n = 7, p=0.003, paired t-test). The extent of depression

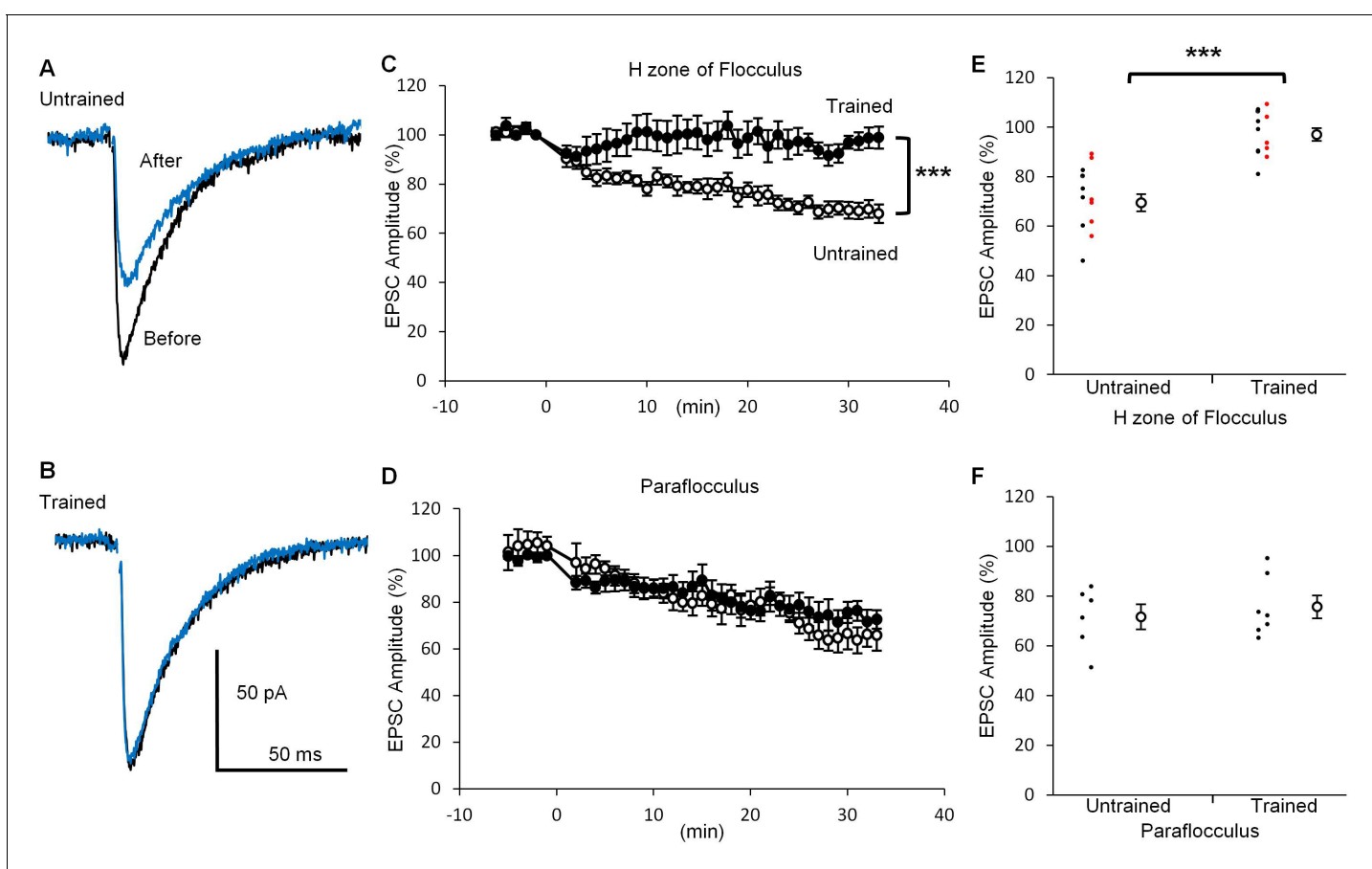

**Figure 3.** OKR training suppressed LTD induction. (A, B) Representative EPSC traces recorded before (black) and 30 min after (blue) the conditioning stimulation in the H zone of untrained (A) or trained (B) mice. (C, D) Time courses of normalized EPSC amplitudes before and after the conditioning stimulation at PF-PC synapse in the H zone (untrained, open circles, n = 14; trained, filled circles, n = 12) (C) or the paraflocculus (untrained, open circles, n = 6; trained, filled circles, n = 7) (D). EPSC amplitudes were normalized by setting the amplitude between −1 min and 0 min at 100%. At 0 min, the conditioning depolarizations coupled with the PF stimulation were applied to a PC. (E, F) Distribution of EPSC amplitudes at 30 min after the start of conditioning relative to those before in the H zone (E) and in the paraflocculus (F). Each small circle represents a result obtained from a PC (red, blind experiment; black, non-blind experiment), and large circles and bars represent Mean ± SEM of all data. ***p<0.001, Student's t-test at 30 min.
DOI: https://doi.org/10.7554/eLife.36209.006

The following source data is available for figure 3:

**Source data 1.** EPSC amplitudes before and after the conditioning stimulation with or without OKR training in the H zone of flocculus or in the paraflocculus.
DOI: https://doi.org/10.7554/eLife.36209.007

was not significantly different between the trained and the untrained mice (p=0.57, Student's t-test). The LTD protocol used here was rather strong one which likely to induce LTD reliably (*Nakamura and Hirano, 2016*). These results are in line with the idea that LTD occurs in the H zone during OKR adaptation, resulting in the decrease in qEPSC amplitude and also in occlusion of additional LTD induction.

### Similar paired-pulse ratio between trained and untrained mice

Finally, we examined whether presynaptic properties at PF-PC synapses were affected by the continuous optokinetic stimulation or not, in order to assess whether postsynaptic properties were specifically changed by the OKR training. The paired pulse ratio (PPR) of EPSC amplitudes reflects the presynaptic release probability (*Debanne et al., 1996*; *Hashimoto and Kano, 1998*), and PF-PC synapses show paired pulse facilitation in clear contrast to climbing fiber-PC synapses showing paired pulse depression (*Konnerth et al., 1990*). The OKR training did not affect PPR with an interval of 100 ms at PF-PC synapses in the H zone (untrained, 1.6 ± 0.0, n = 5; trained, 1.6 ± 0.1, n = 6, p=0.97, Student's t-test), supporting that the presynaptic release probability at PF-PC synapses was not affected by the OKR training (*Figure 4*).

## Discussion

The present results suggest that LTD occurs during OKR adaptation and support the involvement of LTD in motor learning. Previous study reported that OKR training reduced the number of AMPA-type glutamate receptors at PF-PC synapses in the H zone (*Wang et al., 2014*). We think that the decrease in qEPSC amplitudes by the OKR training is functional confirmation of the above morphological results.

Clear suppression of LTD in the floccular PC after OKR training was more than we had expected, although we observed weak LTD in a few cases. It was somewhat unexpected because flocculus receives not only visual inputs, but also vestibular inputs and efference copy signals (*Inagaki and Hirata, 2017*), and during the OKR training there is no vestibular inputs. Thus, the PF-PC synapses

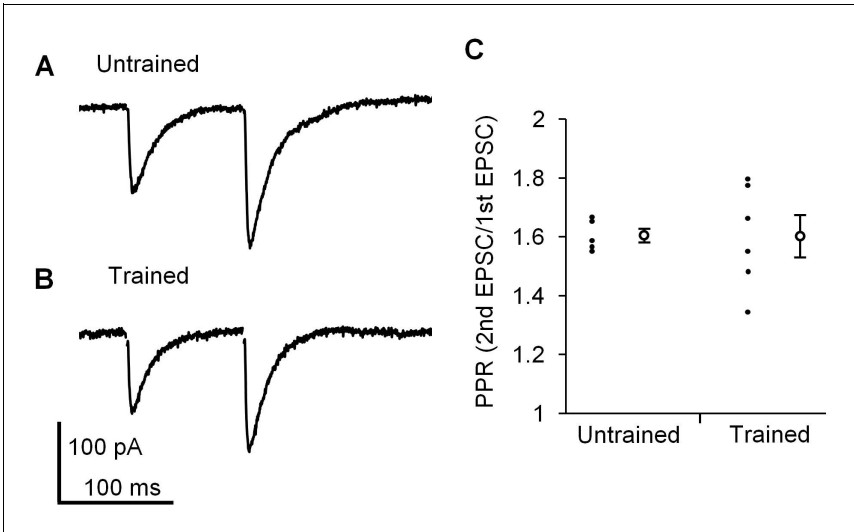

**Figure 4.** Paired pulse facilitation at PF-PC synapse in the H zone. (**A, B**) Representative EPSCs induced by paired-pulse stimulation in the H zone of untrained (**A**) or trained mice (**B**). (**C**) Distribution of paired-pulse ratio of EPSCs with 100 ms interval. Each small circle represents a result obtained from a PC, and large circles and bars represent Mean ± SEM of all data.

DOI: https://doi.org/10.7554/eLife.36209.008

The following source data is available for figure 4:

**Source data 1.** Paired pulse ratio of EPSC amplitudes with or without training in the H zone of flocculus.

DOI: https://doi.org/10.7554/eLife.36209.009

transmitting vestibular signals were unlikely to undergo LTD depending on the PF activity. There might be some mechanisms which allow spreading of depression to the PF synapses transmitting vestibular signals, or the number of the PF inputs conveying vestibular signals is relatively small in the molecular layer region where we applied the PF stimulation. Spread of LTD to inactive PF-PC synapses has been reported (*Wang et al., 2000*; *Reynolds and Hartell, 2001*).

Near complete suppression of LTD (about 20% depression) and relatively small decrease (10%) in qEPSC amplitudes after OKR training may appear contradictory. We think that this apparent discrepancy was caused by larger overestimation of qEPSC amplitudes after LTD occurrence. Because of significant noise in the current traces only qEPSCs larger than 6 pA were analyzed. LTD should have increased the proportion of small qEPSCs, which were not reflected in calculation of mean amplitudes. Thus, qEPSC amplitudes should have been overestimated more after LTD occurrence than before, which must have resulted in underestimation of training effect on qEPSC amplitudes.

LTD is not an only plastic mechanism in the cerebellum, but other synaptic plasticity mechanisms could play roles in motor learning (*Boyden et al., 2004*; *Dean et al., 2010*; *Gao et al., 2012*; *Hirano, 2013*). For example, at PF-PC synapse long-term potentiation (LTP) also occurs, which contributes to motor learning (*Grasselli and Hansel, 2014*). Inhibitory synapses on a PC undergo plasticity (*Kano et al., 1992*; *Kawaguchi and Hirano, 2002*; *He et al., 2015*). Rebound potentiation (RP), which is LTP of GABAergic synaptic transmission on a PC induced by postsynaptic depolarization, is involved in adaptation of vestibulo-ocular reflex, another type of reflex eye movement (*Tanaka et al., 2013*). Thus, multiple synaptic plasticity mechanisms seem to contribute to motor learning. In certain cases, synaptic plasticity mechanisms other than LTD might compensate LTD impairment, and some transgenic mice with specific-defects in LTD might have shown apparently normal learning (*Schonewille et al., 2011*). The present study provides evidence suggesting that LTD does occur during a certain type of motor learning process. Whether other motor learning paradigms also cause LTD or not, and whether other cerebellar plasticity occurs during motor learning are important questions to be addressed in future.

## Materials and methods

### Animals

8–10 weeks old male C57BL/6 mice were used for OKR and electrophysiological recordings. Experimental procedures were carried out in accordance with the guidelines laid down by the National Institutes of Health of the USA and by Kyoto University, and approved by the local committee for handling experimental animals in the Graduate School of Science, Kyoto University (approval number, H2901 and H2902).

### Eye movement recording

The methods for OKR recording were similar to those described previously (*Tanaka et al., 2013*; *Wakita et al., 2017*). A male mouse (8–10 weeks old) was anesthetized with a mixture of 0.9% ketamine and 0.2% xylazine, and a head holder was attached to the skull with small screws using dental cement. The recording was performed 2 days after the surgery. A mouse was fixed on a table surrounded by a cylindrical screen (diameter 30 cm, custom made) with vertical black and white stripes (14°). Sinusoidal horizontal oscillation of the surrounding screen in the light at 1 Hz and 10°/sec peak velocity, was applied to induce OKR. To monitor eye movement, the right eye was illuminated by an infrared LED (TLN201, Toshiba), and the frontal image reflected by a hot mirror (DMR, Kenko) was recorded using an infrared-sensitive CCD camera (XC-HR50, Sony). Because the mirror reflects infrared light but transmits visible light, the eye position was monitored without disrupting the mouse's view. The eye image was captured at 200 Hz, and the eye position was analyzed using a software (Geteye, Morita, Kyoto, Japan) that calculated the pupil centroid. OKR performance was evaluated by measuring the gain. Eye position traces without rapid eye movement or eye blinking during each 50 s recording period was used to estimate the OKR gain. At least 10 cycles of eye position curve were averaged and fitted with a sinusoidal curve by a least square method. The gain was defined as the amplitude of fitted sinusoidal eye position curve divided by that of the screen position. The OKR training was 60 cycles of 50 s optokinetic stimulation followed by 10 s resting period in the light.

## Electrophysiology

The 150 μm coronal slices of the flocculus and paraflocculus of cerebellum were prepared from 8 to 10 weeks old mice < 30 min after the OKR training or without the training, and maintained in Krebs' solution containing the following (in mM): 124 NaCl, 1.8 KCl, 1.2 $KH_2PO_4$, 1.3 $MgCl_2$, 2.5 $CaCl_2$, 26 $NaHCO_3$, and 10 glucose saturated with 95% $O_2$ and 5% $CO_2$ at room temperature (22–24°C). It was necessary to use thin slices to reliably identify H zone of the flocculus. We performed some blind experiments in which electrophysiological recordings and analyses were carried out without knowing whether the mouse was trained or not. The H zone of flocculus was morphologically identified (*Wang et al., 2014*). A PC was whole-cell voltage-clamped at −80 mV with a glass pipette filled with an internal solution containing the following (in mM): 150 CsCl, 0.5 EGTA, 10 HEPES, 2 Mg-ATP, 0.2 Na-GTP and 15 sucrose titrated to pH 7.3 with CsOH. Input resistance (>100 MΩ) and series resistance (10–25 MΩ) were monitored throughout the experiment by applying a voltage pulse (100 ms, −10 mV) every 1 min, and the experiment was terminated when either input or series resistance changed by >20%. EPSCs were recorded in the presence of 20 μM (+)-bicuculline (Merck, Darmstadt), a $GABA_AR$ antagonist. PF-PC EPSC was evoked by applying a 200 μs voltage pulse through a glass electrode containing Krebs' solution which was placed in the molecular layer at 0.05 Hz. To induce LTD a 50 ms voltage pulse to 0 mV coupled with a PF stimulation at 15 ms after the onset of depolarization, was applied 10 times at 1 Hz. qEPSC was recorded in Krebs' solution in which $CaCl_2$ was replaced with $SrCl_2$, and voltage pulse to evoke PF-EPSC was applied 60 μm away from the soma of PC. Asynchronous EPSCs occurring 150–950 ms after the PF stimulation with appropriate time courses and amplitudes > 6 pA were selected as qEPSCs, and analyzed with Mini Analysis software (Synaptosoft, RRID: SCR_014441). Noisy data in which detection of 6 pA qEPSC was difficult were excluded from analysis. The mean qEPSC amplitude was calculated from >100 events in a PC. The mean of paired-pulse ratio of EPSC amplitudes was calculated from five trials in a PC.

## Statistics

Data are presented as mean ± SEM. Sample sizes were determined based on previous publications on the cerebellar synaptic plasticity (*Aiba et al., 1994*; *De Zeeuw et al., 1998*; *Linden, 2001*; *Hirai et al., 2005*; *Hansel et al., 2006*; *Takeuchi et al., 2008*; *Schonewille et al., 2011*; *Tanaka et al., 2013*; *He et al., 2015*; *Yamaguchi et al., 2016*). The normality of data was examined with Kolmogorov-Smirnov test. The two-tailed paired t-test was used to compare the mean value of OKR gain. The two-tailed Student's t-test was used to compare EPSC amplitudes or PPR. α value was set at 0.05.

## Acknowledgements

This work was supported by a grant 17H05566 from the Ministry of Education, Culture, Sports, Science and Technology in Japan to TH and a grant 17J07402 from the Japan Society for the Promotion of Science to T I. We thank Drs. Y Tagawa, H Tanaka, G Ohtsuki and S Kawaguchi for comments on the manuscript.

## Additional information

### Funding

| Funder | Grant reference number | Author |
| --- | --- | --- |
| Ministry of Education, Culture, Sports, Science, and Technology | 17H05566 | Tomoo Hirano |
| Japan Society for the Promotion of Science London | 17J07402 | Takuma Inoshita |

The funders had no role in study design, data collection and interpretation, or the decision to submit the work for publication.

## Author contributions
Takuma Inoshita, Conceptualization, Data curation, Formal Analysis, Funding acquisition, Investigation, Methodology, Writing—original draft; Tomoo Hirano, Conceptualization, Supervision, Funding acquisition, Validation, Investigation, Writing—original draft

## Author ORCIDs
Tomoo Hirano (iD) http://orcid.org/0000-0003-3685-5935

## Ethics
Animal experimentation: Experimental procedures were carried out in accordance with the guidelines laid down by the National Institutes of Health of the USA and by Kyoto University, and approved by the local committee for handling experimental animals in the Graduate School of Science, Kyoto University (approval number, H2901 and H2902).

## Decision letter and Author response
Decision letter https://doi.org/10.7554/eLife.36209.012
Author response https://doi.org/10.7554/eLife.36209.013

## Additional files

### Supplementary files
• Transparent reporting form
DOI: https://doi.org/10.7554/eLife.36209.010

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
