## [Decision Letter]

[Editors’ note: a previous version of this study was rejected after peer review, but the authors submitted for reconsideration. The first decision letter after peer review is shown below.]

Thank you for submitting your work entitled "Occurrence of long-term depression in the cerebellar flocculus during adaptation of optokinetic response" for consideration by *eLife*. Your article has been evaluated by a Senior Editor and three reviewers, one of whom, Jennifer L Raymond (Reviewer #3), is a member of our Board of Reviewing Editors.

Our decision has been reached after consultation between the reviewers. Based on these discussions and the individual reviews below, we regret to inform you that your work was deemed too preliminary at this point for publication in *eLife*. However, the reviewers did find the research to have considerable potential impact, hence we would be willing to consider a revised version of the manuscript with a more thoroughly analyzed and convincing data set.

The experiments described represent a potentially important milestone for the cerebellar field, since this is the first study to use ex-vivo experiments (which deliver behavioral training then study the synaptic correlates in slice) to study the induction of LTD in the cerebellum during learning. The results have the potential to solidify the evidence for a role of cerebellar LTD in OKR adaptation, providing a key type of evidence that was previously lacking. Moreover, the establishment of this approach could set a new standard for studying the role of LTD in other forms of cerebellum-dependent learning, for which the role of LTD is more controversial. However, to accomplish these goals, the reviewers felt that the data must be more convincing than presented.

Most importantly, the reviewers found the qEPSC results to be not sufficiently convincing. Additional experiments should be conducted with the physiologist blind to the behavioral training condition experienced by the animal. Moreover, the data should be presented in a more informative manner, showing the distributions of the individual data sets, not just the average value.

Also, the discrepancy between the size of the effect of training on the qEPSC vs. the size of the LTD that is occluded merits additional discussion. The near-complete occlusion of LTD after behavioral training is very surprising, for the reasons indicated by the authors, and also because the normal LTD amplitude is ~20%, but the reduction in qEPSC amplitude in the trained animals was relatively small (<10%). Potential reasons for this discrepancy should be discussed, and put into the context of previous work in the field.

A more minor issue is that the authors should justify their choice of LTD induction protocol. Since it is on the "strong" end of LTD induction protocols that are used, this may help to make the case that LTD is really occluded, even when they hit it with a strong induction protocol. But if that was the rationale for this choice, it is not stated in the manuscript.

The initial complete reviews are appended for your consideration, but the most essential issues are described above. *eLife*

*Reviewer #1:*

In the present manuscript, Inoshita and Hirano present evidence that LTD occurs in the cerebellar flocculus after OKR training in mice. The authors show that amplitudes of quantal EPSCs (qEPSCs) in slices prepared after training are reduced and that LTD induction is occluded in slices from trained animals. While the findings are certainly of interest, the impact of the data is limited by small effect size (qEPSC measures). Neither the demonstration of LTD after cerebellar training (see work from Cheron and colleagues), nor the study of LTD in the flocculus (see work from Raymond and colleagues) is particularly novel. Furthermore, the study is entirely descriptive and does not address the question whether LTD contributes to OKR learning.

The difference in qEPSC amplitudes between recordings from trained and untrained mice is very small (less than 10 percent). This small effect size is a problem as it is not a trivial task to quantify qEPSC measures and the traces shown do not make a convincing case. The low effect amplitude is dramatically smaller than the LTD amplitude that is typically reported. There might be reasons for this (e.g. intact inhibition during the induction phase in the intact animal), but this is not discussed. An additional means to present the data in a more informative way would be to show the spread of individual data sets, not just the average value.

*Reviewer #2:*

In this short report, the authors studied the correlation between long term depression (LTD) at the granule cell to Purkinje cell synapses (GC-PC) and the adaptation of the optokinetic reflex (OKR), a well described model of motor learning. They showed that the gain of the OKR can be increased and correlate with a decrease in the amplitude of the quantal EPSCs recorded in brain slices at the GC-PC synapses in the H zone of the flocculus, suggesting that LTD occurred at these synapses. LTD induction was occluded by OKR training protocol without any change in paired pulse facilitation. They conclude that postsynaptic LTD occurs at the GC-PC synapses during motor learning.

My major concern is that the main result of this manuscript is not really new and more importantly does not bring new hypotheses about the link between LTD and motor learning. Several articles already correlated LTD and OKR adaptation in mice (not only VOR) in the past (Shutoh F et al. Neurosci Res. 2002 Feb;42(2):141-5 PMID: 11849733 and Eur J Neurosci 2003 18:134-42 PMID 12859346; Endo S et al. PNAS 2009 Mar 3;106(9):3525-30.PMID:19218432; Wang et al., 2014). Also, occlusion experiments by plasticity induction have been already recently demonstrated in the flocculus (Nguyen-Vu et al., *eLife*2017).

I agree with the author that the subject appears controversial: some studies demonstrated that LTD is important for VOR/OKR adaptation, mostly in the gain increase protocols, while others showed that LTD is not mandatory, specifically for gain decrease protocols. But it turns out that these discrepancies have different origins (for review, Ito et al. Prog Brain Res 2014; 210:1-30 PMID 24916287; Boyden et al., 2004; De Zeeuw and Ten Brinke, CSH Perspect Biol 2015;7:a021683). For example, different types of plasticity may occur to compensate for the lack of LTD, masking the physiological role of LTD (Gao et al., 2012). Also, LTD protocols varied a lot and LTD induction is highly sensitive to different features (timing, number of pulses, internal solution, depolarization) suggesting that a lack of LTD for a given protocol does not necessary mean that LTD is abolished for another one (see Yamaguchi et al., 2016).

In this article, the authors used a non-physiological induction protocol (10 depolarizations of the Purkinje cell while PF are stimulated at 1 Hz and a highly permissive internal solution with Cesium ions) and without CF stimulation, which might lead to the activation of a specific transduction pathway. Finally, although it was the main issue, no causal or direct link between learning mechanisms and LTD can be inferred from this study. Therefore, I don't think that these findings can be considered of very high importance.

Other comments:

- In the quantal analysis, the histogram of the distribution of individual events should be shown in order to demonstrate that the threshold (set at 6 pA) for detection is well set and that a Gaussian distribution is actually observed. Also, knowing the distribution of the mean qEPSC (see Figure 2—source data 1), sample size effects might have influenced the results.

- Paired Pulse frequency is not mentioned.

*Reviewer #3:*

This manuscript provides evidence that LTD is induced at the cerebellar parallel fiber-Purkinje cell (pf-Pk LTD) synapses after adaptation of the optokinetic reflex (OKR). Although there was already considerable evidence for a role of pf-Pk LTD in this form of cerebellum-dependent learning, the evidence provided in this manuscript is some of the most direct-showing a decrease in unitary EPSC amplitude in slices from trained vs. untrained animals, plus occlusion of LTD in slices from the trained animals. The results solidify the evidence for a role of pf-Pk LTD in OKR adaptation, and establish a new approach that could be used to study the role of LTD in other forms of cerebellum-dependent learning, for which the role of LTD is more controversial.

The evidence here is consistent with and complementary to previous evidence for a role of LTD in OKR adaptation.

The near-complete occlusion of LTD after behavioral training is very surprising, for the reasons indicated by the authors, and also because the normal LTD amplitude is ~20%, but the reduction in qEPSC amplitude in the trained animals was only ~10%. This represents an interesting new insight, which may be related to other work in the field on the occlusion of plasticity (Nguyen-Vu et al., *eLife*2017) and the spread of LTD to nearby synapses (Wang et al. 2000; Reynolds and Hartell, 2001).

This type of ex-vivo experiment, which delivers behavioral training, then studies the cellular correlates in slice, has rarely been used in the cerebellum (the one study I can think of is Schreurs et al., 1997). Hence, one of the main merits of the current work is providing a new standard for establishing a contribution of pf-Pk LTD to learning, which can be used to assess the role of LTD in other forms of cerebellum-dependent learning. It was not necessarily the case that LTD could be detected in ex-vivo slices, since it might have been washed out in the slicing process.

For this type of experiment, it is optimal that the physiologist is blind to the behavioral condition of the animal, and that does not seem to be the case here.

The authors measured input resistance and series resistance, and used changes in these measures during the experiment as a criterion for exclusion. Were these measures similar in trained and untrained animals? These, and any other measures of the basic physiology that were made, should be provided.

---

## [Author Response]

[Editors’ note: the author responses to the first round of peer review follow.]

[…] Most importantly, the reviewers found the qEPSC results to be not sufficiently convincing. Additional experiments should be conducted with the physiologist blind to the behavioral training condition experienced by the animal. Moreover, the data should be presented in a more informative manner, showing the distributions of the individual data sets, not just the average value.

We have performed blind experiments on qEPSC measurement and LTD recording in the H-zone of flocculus in which we confirmed our previous results and the data have been presented in more informative manner showing all individual values in Figure 2 and 3.

Also, the discrepancy between the size of the effect of training on the qEPSC vs. the size of the LTD that is occluded merits additional discussion. The near-complete occlusion of LTD after behavioral training is very surprising, for the reasons indicated by the authors, and also because the normal LTD amplitude is ~20%, but the reduction in qEPSC amplitude in the trained animals was relatively small (<10%). Potential reasons for this discrepancy should be discussed, and put into the context of previous work in the field.

We thank the editors and reviewers for this comment. Admittedly, our presentation of data and explanation were insufficient. We have shown histograms of qEPSC amplitudes in Figure 2 and added explanations (subsection “Smaller unitary excitatory postsynaptic response in the H zone of OKR trained mice” and Discussion, second paragraph). Because of the significant noise in current traces only qEPSCs larger than 6 pA were measured. Thus, smaller qEPSCs were not considered. LTD should have increased the proportion of small qEPSCs, which could not be analyzed and were not reflected in calculation of mean amplitudes. Thus, qEPSC amplitudes should have been overestimated more after LTD occurrence than before, which must have resulted in underestimation of training effect on qEPSC amplitudes.

A more minor issue is that the authors should justify their choice of LTD induction protocol. Since it is on the "strong" end of LTD induction protocols that are used, this may help to make the case that LTD is really occluded, even when they hit it with a strong induction protocol. But if that was the rationale for this choice, it is not stated in the manuscript.

Thank you for the comment again. It was exactly the case. We have added explanation in the text (subsection “Suppression of LTD induction in the H zone of trained mice”).

The initial complete reviews are appended for your consideration, but the most essential issues are described above.Reviewer #1:In the present manuscript, Inoshita and Hirano present evidence that LTD occurs in the cerebellar flocculus after OKR training in mice. The authors show that amplitudes of quantal EPSCs (qEPSCs) in slices prepared after training are reduced and that LTD induction is occluded in slices from trained animals. While the findings are certainly of interest, the impact of the data is limited by small effect size (qEPSC measures). Neither the demonstration of LTD after cerebellar training (see work from Cheron and colleagues), nor the study of LTD in the flocculus (see work from Raymond and colleagues) is particularly novel. Furthermore, the study is entirely descriptive and does not address the question whether LTD contributes to OKR learning.

We thank the reviewer for the interest in our study. We have rewritten the Introduction and added explanation to better clarify the limitation and significance of this study (Introduction, second paragraph).

The difference in qEPSC amplitudes between recordings from trained and untrained mice is very small (less than 10 percent). This small effect size is a problem as it is not a trivial task to quantify qEPSC measures and the traces shown do not make a convincing case. The low effect amplitude is dramatically smaller than the LTD amplitude that is typically reported. There might be reasons for this (e.g. intact inhibition during the induction phase in the intact animal), but this is not discussed. An additional means to present the data in a more informative way would be to show the spread of individual data sets, not just the average value.

We thank the reviewer for the comments. Admittedly, our presentation of data and explanation were insufficient. We have added histograms of qEPSC amplitudes in Figure 2 and added explanation (subsection “Smaller unitary excitatory postsynaptic response in the H zone of OKR trained mice” and Discussion, second paragraph). Because of the significant noise in current traces only qEPSCs larger than 6 pA were analyzed. Thus, smaller qEPSCs were not considered. Skewed distributions of qEPSC amplitudes shown in Figure 2 suggest that there should have been smaller qEPSCs which were not analyzed, and LTD occurrence should have increased the proportion of small qEPSCs which were not reflected in calculation of mean amplitudes. Thus, qEPSC amplitudes should have been overestimated more after LTD occurrence than before, which must have resulted in underestimation of training effect on qEPSC amplitudes. We have also presented each mean of qEPSC amplitudes in Figure 2.

Reviewer #2:[…] My major concern is that the main result of this manuscript is not really new and more importantly does not bring new hypotheses about the link between LTD and motor learning. Several articles already correlated LTD and OKR adaptation in mice (not only VOR) in the past (Shutoh F et al. Neurosci Res. 2002 Feb;42(2):141-5 PMID: 11849733 and Eur J Neurosci 2003 18:134-42 PMID 12859346; Endo S et al. PNAS 2009 Mar 3;106(9):3525-30.PMID:19218432; Wang et al., 2014). Also, occlusion experiments by plasticity induction have been already recently demonstrated in the flocculus (Nguyen-Vu et al., 2017).I agree with the author that the subject appears controversial: some studies demonstrated that LTD is important for VOR/OKR adaptation, mostly in the gain increase protocols, while others showed that LTD is not mandatory, specifically for gain decrease protocols. But it turns out that these discrepancies have different origins (for review, Ito et al. Prog Brain Res 2014; 210:1-30 PMID 24916287; Boyden et al., 2004; De Zeeuw and Ten Brinke, CSH Perspect Biol 2015;7:a021683). For example, different types of plasticity may occur to compensate for the lack of LTD, masking the physiological role of LTD (Gao et al., 2012).

Important new information we report here is the decrease in qEPSC amplitudes after OKR training, and the evidence whether LTD occurs or not during motor learning in wild-type animal has not been complete. We think we have provided an experimental support for the occurrence of LTD during OKR adaptation. We have added explanations about previous relevant papers and tried to better clarify the limitation and significance of this study in Introduction.

Also, LTD protocols varied a lot and LTD induction is highly sensitive to different features (timing, number of pulses, internal solution, depolarization) suggesting that a lack of LTD for a given protocol does not necessary mean that LTD is abolished for another one (see Yamaguchi et al., 2016).In this article, the authors used a non-physiological induction protocol (10 depolarizations of the Purkinje cell while PF are stimulated at 1 Hz and a highly permissive internal solution with Cesium ions) and without CF stimulation, which might lead to the activation of a specific transduction pathway. Finally, although it was the main issue, no causal or direct link between learning mechanisms and LTD can be inferred from this study. Therefore, I don't think that these findings can be considered of very high importance.

In this study, we used strong LTD induction protocol to examine occlusion of LTD occurrence reliably. Explanation of this reasoning has been added in the text (subsection “Suppression of LTD induction in the H zone of trained mice”).

Other comments:- In the quantal analysis, the histogram of the distribution of individual events should be shown in order to demonstrate that the threshold (set at 6 pA) for detection is well set and that a Gaussian distribution is actually observed. Also, knowing the distribution of the mean qEPSC (see Figure 2—source data 1), sample size effects might have influenced the results.

We thank the reviewer for this comment. Admittedly, our presentation of data and explanation were insufficient. We have added histograms of qEPSC amplitudes in Figure 2 and added explanation (subsection “Smaller unitary excitatory postsynaptic response in the H zone of OKR trained mice” and Discussion, second paragraph). Because of the significant noise in current traces only qEPSCs larger than 6 pA were counted. Thus, smaller qEPSCs were not considered here. LTD should have increased the proportion of small qEPSCs, which were not reflected in calculation of mean amplitudes. Thus, qEPSC amplitudes should have been overestimated more after LTD occurrence than before, which must have resulted in underestimation of training effect on qEPSC amplitudes. We have also added results of blind experiments and presented each mean of qEPSC amplitudes in Figure 2.

- Paired Pulse frequency is not mentioned.

100 ms interval. We have added explanation (subsection “Similar paired-pulse ratio between trained and untrained mice”).

Reviewer #3:[…] The near-complete occlusion of LTD after behavioral training is very surprising, for the reasons indicated by the authors, and also because the normal LTD amplitude is ~20%, but the reduction in qEPSC amplitude in the trained animals was only ~10%. This represents an interesting new insight, which may be related to other work in the field on the occlusion of plasticity (Nguyen-Vu et al., 2017) and the spread of LTD to nearby synapses (Wang et al. 2000; Reynolds and Hartell, 2001).This type of ex-vivo experiment, which delivers behavioral training, then studies the cellular correlates in slice, has rarely been used in the cerebellum (the one study I can think of is Schreurs et al., 1997). Hence, one of the main merits of the current work is providing a new standard for establishing a contribution of pf-Pk LTD to learning, which can be used to assess the role of LTD in other forms of cerebellum-dependent learning. It was not necessarily the case that LTD could be detected in ex-vivo slices, since it might have been washed out in the slicing process.

We thank the reviewer for this comment. Admittedly, our presentation of data and explanation were insufficient. We have added histograms of qEPSC amplitudes in Figure 2 and added explanation (Introduction, second paragraph, subsection “Smaller unitary excitatory postsynaptic response in the H zone of OKR trained mice” and Discussion, second paragraph). Because of the significant noise in current traces only qEPSCs larger than 6 pA were measured. Thus, smaller qEPSCs were not considered here. LTD should have increased the proportion of small qEPSCs, which were not reflected in calculation of mean amplitudes. Thus, qEPSC amplitudes should have been overestimated more after LTD than before, which must have resulted in underestimation of training effect on qEPSC amplitudes. We have also added results of blind experiments and presented each mean of qEPSC amplitudes obtained from a Purkinje cell in Figure 2. We have rewritten Introduction and added explanation of previous relevant reports, and tried to better clarify the significance of this study.

For this type of experiment, it is optimal that the physiologist is blind to the behavioral condition of the animal, and that does not seem to be the case here.

We performed blind experiments on qEPSC measurement and LTD induction in the H-zone of flocculus, which confirmed our previous results. The results have been presented in Figure 2 and Figure 3, and explanations have been added (subsection “Smaller unitary excitatory postsynaptic response in the H zone of OKR trained mice”, subsection “Suppression of LTD induction in the H zone of trained mice” and subsection “Electrophysiology”).

The authors measured input resistance and series resistance, and used changes in these measures during the experiment as a criterion for exclusion. Were these measures similar in trained and untrained animals? These, and any other measures of the basic physiology that were made, should be provided.

The data have been presented in Figure 2, and explanation has been added (subsection “Smaller unitary excitatory postsynaptic response in the H zone of OKR trained mice”).